# Deep Learning versus Spectral Techniques for Frequency Estimation of Single Tones: Reduced Complexity for Software-Defined Radio and IoT Sensor Communications

**DOI:** 10.3390/s21082729

**Published:** 2021-04-13

**Authors:** Hind R. Almayyali, Zahir M. Hussain

**Affiliations:** 1Computer Science and Mathematics, University of Kufa, Najaf 54001, Iraq; hindalmayyali@gmail.com; 2School of Engineering, Edith Cowan University, Joondalup, WA 6027, Australia

**Keywords:** frequency estimation, deep-learning (DL), sensors, Internet of Things (IoT), short word length (SWL), software-defined radio (SDR), parallel-computing fast Fourier transform (FFT), low-power, low-cost, biomedical sensors

## Abstract

Despite the increasing role of machine learning in various fields, very few works considered artificial intelligence for frequency estimation (FE). This work presents comprehensive analysis of a deep-learning (DL) approach for frequency estimation of single tones. A DL network with two layers having a few nodes can estimate frequency more accurately than well-known classical techniques can. While filling the gap in the existing literature, the study is comprehensive, analyzing errors under different signal-to-noise ratios (SNRs), numbers of nodes, and numbers of input samples under missing SNR information. DL-based FE is not significantly affected by SNR bias or number of nodes. A DL-based approach can properly work using a minimal number of input nodes N at which classical methods fail. DL could use as few as two layers while having two or three nodes for each, with the complexity of O{N} compared with discrete Fourier transform (DFT)-based FE with O{Nlog2 (N)} complexity. Furthermore, less N is required for DL. Therefore, DL can significantly reduce FE complexity, memory cost, and power consumption, which is attractive for resource-limited systems such as some Internet of Things (IoT) sensor applications. Reduced complexity also opens the door for hardware-efficient implementation using short-word-length (SWL) or time-efficient software-defined radio (SDR) communications.

## 1. Introduction

Estimating the frequency of a single-tone sinusoidal wave under noise is a fundamental problem in signal processing for communications, and its effect is extended to biomedical and power engineering [1,2].

Several classical methods were proposed to estimate the frequency of a single-tone sinusoid, mostly based on Fourier analysis [3].

Correlation methods can be very attractive for hardware implementation, IoT, sensors, and software-defined ratio (SDR), as they are much more computationally efficient than spectral techniques are. However, those techniques are far less accurate compared to Fourier-based techniques, especially under low values of signal-to-noise ratio (SNR) [4]. 

Phase-locked loops (PLLs) are widely used in communication systems to handle this problem. However, PLLs can be slower than spectral or correlative techniques as they need time for locking [5]. More samples are needed in digital PLLs (DPLLs) if more realizations are considered for better results, especially under a low SNR where noise significantly disturbs the locking process.

Recently, machine learning has emerged as a powerful tool for various tasks in many fields, including communications and signal processing [6,7,8]. Despite the increased attention to machine learning, very few works considered its application to handle the problem of frequency estimation. 

In [9], an approach based on deep learning for single-tone frequency estimation under noise was proposed. Unlike Fourier-based, PLL, or correlative techniques, the DL-based approach requires prior knowledge of the SNR. However, SNR estimation can be in error under varying noise conditions; therefore, many studies, including [9], assumed that SNR is constant during the period of frequency estimation. 

In this work, we further investigate the performance DL for frequency estimation under insufficient SNR information. On the other hand, the effect of the number of nodes was also studied, and no significant change in performance was obtained after increasing the number of nodes in hidden layers. The contribution of this work has the following directions:The performance of the DL approach is compared with the performance of still-active classical techniques that are based on Fourier analysis.In many situations, SNR estimation can be inaccurate or unavailable. Hence, system performance is investigated in case of unavailable SNR information.The DL approach is SNR-dependent; hence, an investigation of system performance under various SNRs is presented.The more the nodes in the DL approach are, the better the accuracy of estimation could be. Hence, this work investigated the effect of changing the number of nodes in the hidden layers of the network.The number of input samples (signal length or duration) has significant impact on the complexity and the performance of classical and DL-based methods. This point was fully investigated in this work.The effect of different realizations while training was handled in the literature of DL-based approaches, as it is a necessary step in the training process. However, the possibility of different realizations in the working environment (application phase) was not previously handled. In this work, we discuss the effect of different realizations during the application phase.The reduced complexity introduced by DL-based FE, in addition to avoiding complex-valued arithmetic operations, makes FE easier and cheaper for IoT communications, sensors, sensor networks, and software-defined radio (SDR). This work presents a discussion on such possibilities.

This paper is organized as follows. Section 2 presents related works; Section 3 presents the signal model and the problem definition; Section 4 reviews the most-effective classical techniques; and Section 5 addresses the yet-unhandled issues of the DL approach and presents the results in a comparative study that reveals the power of DL versus classical frequency-estimation techniques.

## 2. Motivation and Related Work

The problem of frequency estimation (FE) is often handled using classical Fourier and correlative techniques. However, some recent works handled this problem by neural networks and deep learning. Good accuracy was obtained; hence, the use of DL for frequency estimation is promising. 

The authors in [9] designed a three-layer neural network to estimate the frequency of a sinusoid contaminated by a white-noise process at an SNR of 25 dB, where the trained model could estimate the frequency of any previously unseen noisy sinusoid in a short amount of time. However, this work did not address some important issues, including comparisons with classical techniques, system performance under wrong SNR estimation, system performance under various SNR values, and the effect of the number of nodes. Specifically, the following points are not clear in [9]: The network was trained for a specific SNR. There should be a clarification whether different networks should be present in the case that different SNRs are expected.The effect of SNR on estimation error was unclear.The number of input nodes was chosen as N = 2000. It is not clear whether this choice of the input samples (nodes) is frequency-, duration-, or network-dependent.The division of the frequency range was not clear. The effect of this division on estimation error should be addressed. The relation of this division to the time–frequency uncertainty principle should also be clarified.The number of nodes in the hidden layers was 2. The effect of this number on estimation accuracy should be clarified. The possibility that this effect is SNR-dependent should be addressed.Classical techniques for frequency estimation have been well-studied for decades. There should be a clear reasoning why one should choose DL-based estimation instead.

In [10], the authors presented an approach based on deep neural networks to estimate the fundamental frequency (F0), which is an essential acoustical feature to find the audible pitch level. This problem is classically dealt with using time-domain or correlative methods; however, the authors in [5] applied a different direction via deep learning where error was reduced as compared with classical techniques.

In [11], the authors presented an approach for estimating the frequency of linear frequency modulation (LFM) signals, a topic that has applications in radar and communication engineering. The approach utilized a convolutional neural network (CNN), while this problem was traditionally handled using time–frequency analysis [12]; however, this approach requires significant computational cost as it involves two-dimensional transforms.

## 3. Problem Statement

The single-tone sinusoidal generic model is given by
(1)$$s\left(t\right)\;=\;a\left(t\right)\cos\left(2\pi f_{o}t+\theta_{o}\right),\;$$
where a(t) is amplitude; normally, the maximal frequency in its spectrum is much lower than sinusoidal frequency $f_{o}$ (i.e., slowly varying). $\theta_{o}$ is the initial phase. 

As a carrier for information signals in communication systems via modulation, the sinusoidal carrier normally has constant amplitude, while the receiving device can apply amplitude estimation and a constant-gain multiplier to restore the original amplitude. Therefore, here, we consider sinusoidal model
(2)$$s\left(t\right)\;=\;A\;\cos\left(2\pi f_{o}t+\theta_{o}\right)$$
with constant amplitude A.

The signal in Equation (2) is transmitted as a pilot to allow for the receiver to estimate the carrier frequency. However, such a single-tone signal can undergo frequency changes due to the Doppler effect and oscillator instabilities (see [1] and the references therein). Even minor change in carrier can cause demodulation problems. The performance of Orthogonal Frequency Division Multiplexing (OFDM) systems can deteriorate for improper carrier estimation [1]. Hence, accurate frequency estimation of the incoming carrier is very important for correct demodulation at the receiver. 

However, communication channels add noise to transmitted signals, making frequency estimation a more challenging task. Therefore, the actual carrier model used in frequency estimation is as follows.
(3)$$s\left(t\right)\;=\;A\;\cos\left(2\pi f_{o}t+\theta_{o}\right)+\eta \left(t\right),$$
where η(t) represents the noise process. In most communication systems, noise is modelled as a Gaussian (normal) process with zero mean and variance that equals noise power $\sigma^{2}$ and the process is normally referred to as $N\left(0,\sigma^{2}\right)$. 

Signal-to-noise ratio (SNR) is defined as follows.
(4)$$\mathrm{SNR}\;=\;r\;=\;\frac{\mathrm{signal}\;\mathrm{power}}{\mathrm{noise}\;\mathrm{power}}\;=\;\frac{A^{2}/2}{\sigma^{2}}$$
and usually is expressed in decibels:(5)$${\mathrm{SNR}}_{\mathrm{dB}}\;=\;10\;\log_{10}\left(\mathrm{SNR}\right)\;$$
or simply
$$\;r_{\mathrm{dB}}\;=\;10\;\log_{10}\left(r\right),$$
which is more convenient to logarithmically handle large and small values.

This work focuses on the problem of estimating frequency $f_{o}$ of the single-tone model in Equation (3) under Gaussian noise η(t). Our approach was to use a deep-learning (DL) neural network with multiple layers in this process, following recent works that confirmed the high accuracy of this approach [9,10,11]. Despite the pioneering role of these works, they did not address many fundamental issues in DL frequency estimation. We focus on clarifying these issues, and define the research problems as follows.

Comparing the performance of DL-based approach with classical techniques.Investigating system performance in case of unavailable SNR information.Investigating system performance under various SNRs.Investigating the effect of changing the number of nodes in the hidden layers of the network.Investigating the effect of input signal length (duration) on the performance of both classical and DL-based methods. This factor has significant impact on overall performance and complexity.Investigating the effect of different realizations during the application phase (not only during the training phase).Investigating the impact of DL-based FE on IoT, sensors, sensor networks, and software-defined radio (SDR).

In addition to the above, investigating the effect of different realizations of the input signal in the application phase of the DL approach would be handled. It is known that different realizations of any statistical experiment would result in better estimation of its random variable. This point could be treated by increasing the number of input nodes; however, this direction would complicate the network structure and limit the choice of the user who may be satisfied with a user-defined accuracy level. 

The next section presents an overview of main well-known approaches in classical FE for single-tone signals.

## 4. A Brief Overview of Classical FE for Single Tones

Literature surveys of classical frequency-estimation methods are presented in [2,13,14]. In [1], an iterative approach for single-tone frequency estimation was presented; however, despite their benefits in accuracy, such approaches can be computationally expensive or complex. 

In [15,16,17] and the references therein, good DFT interpolators were presented; however, despite some accuracy gains, they were computationally more expensive than earlier methods in the literature were. 

Reduced complexity is a property that can be very useful for some resource-limited systems such as wireless sensor networks [18]. The complexity–accuracy–speed trade-off is the selection criterion in this work. 

Here, a brief overview of the most accurate, computationally efficient classical approaches is presented. All accurate classical FE methods are based on using the discrete Fourier transform (DFT) of the signal:(6)$$S\left(k\right)\;=\;\overset{N-1}{\underset{n\;=\;0}{\sum }}s\left(n\right)\;e^{-j2\pi kn/N}\;$$
where N is the number of input samples. Various methods were designed for FE on the basis of interpolating the DFT of the sinusoid. The most accurate and computationally efficient methods under additive Gaussian noise are as follows.

### 4.1. Maximum of DFT Estimator

This method is based on the location (index) of the maximum of the absolute of the DFT, given by
(7)$$I_{m}\;=\;\arg\left\{\underset{k}{\max}\left[\left|S\left(k\right)\right|\right]\right\}$$
since the sinusoid gives a delta spectrum. Estimated frequency is given by
(8)$$F_{o}\;=\;\frac{I_{m}\cdot f_{s}}{N},$$
where $f_{s}$ is sampling frequency.

The error of estimation at SNR = r (dB) is defined in this paper as
(9)$$e\left(r\right)\;=\;\frac{\left|F_{o}-f_{o}\right|}{f_{o}},\;$$
and the above definition of the error was used to evaluate the performance of DL approach versus existing methods. Another performance criterion is the computational complexity of the method.

### 4.2. Quadratic Interpolator

This approach designs a quadratic interpolation for Fourier transform (DFT) using three points near the index of the maximum of |DFT|, i.e., the following points in the DFT domain:$$\left(I_{m}-1,\;S_{1}\;=\;\left|S_{I_{m}-1}\right|\right),(I_{m},S_{2}\;=\;\left|S_{I_{m}}\right|),(I_{m}+1,S_{3}\;=\;\left|S_{I_{m}+1}\right|),$$
where $I_{m}$ is given in Equation (7). The estimated frequency is given by
(10)$$F_{o}\;=\;\frac{I_{o}\cdot f_{s}}{N},$$
where $I_{o}\;=\;I_{m}+\delta ;$
$\delta \;=\;\left(S_{3}-S_{1}\right)\;/[2*\left(2*S_{2}-S_{1}-S_{3}\right)]$.

### 4.3. Barycentric Estimator

This estimator is similar to the quadratic estimator, with the estimated frequency given by
(11)$$F_{o}\;=\;\frac{I_{b}\cdot f_{s}}{N},$$
where $I_{b}\;=\;I_{m}+\alpha ;$
$\alpha \;=\;\left(S_{3}-S_{1}\right)\;/[S_{1}+S_{2}+S_{3}]$.

During this work, the barycentric estimator gave the best results in terms of error among the selected classical techniques.

The next section presents the structure and training details of the deep-learning network that was adopted for the FE of single-tone signals.

## 5. Deep Learning for Single-Tone FE: Network Structure and Training

In this section, the structure of the DL approach for the frequency estimation of single tones is analyzed and justified. In the DL-based approach, complexity only exists during the training phase and not in the actual network structure. The significant reduction in complexity can make the DL approach for FE suitable for IoT communications involving wireless sensor networks (WSNs).

### 5.1. DL Network Structure

A deep neural network with two hidden layers is shown in Figure 1, where arrows represent the multiplication of relevant items by specific weights and adding biases, using the MATLAB tansig (or tanh) function as the pointwise activation function in hidden layers as follows.
(12)$$\Phi \left\{Z_{N\times 1}\right\}\;=\;\Phi \left\{\left[z_{i}|i\;=\;1:N\right]\right\}\;=\;{\left[\phi (z_{i})\right]}_{N\times 1}\;=\;{\left[\frac{2}{1+e^{-2z_{i}}}-1\right]}_{N\times 1},$$
where N is the number of layer output nodes.

Tanh activation for hidden layers is used to prevent results from expanding in magnitude and normalizing them in the [−1, 1] range. Both positive and negative values are needed for the proper adjustment of the network, unlike the case in the output layer, where a single output (a decision) is obtained using the positive linear activation function (ReLU) to give a proportional value to the input frequency as follows.
(13)$$\vartheta \left(x\right)\;=\;\max\left(0,x\right)\;=\;\left\{\begin{array}{ccc} x & ; & x\geq 0\\ 0 & ; & x<0 \end{array}\right\},$$
where ReLU is used because the actual value of the frequency is needed, knowing that the frequency is always positive.

The operation of the 2-layer network in Figure 1 can be expressed as follows.
(14)$$f\;=\;\vartheta \left[V_{1\times K}\cdot \Psi \left\{H_{K\times J}\cdot \Phi \left(W_{J\times N}\cdot X_{N\times 1}+b_{J\times 1}\right)+\beta_{K\times 1}\right\}+{𝒷}_{1\times 1}\right],\;$$
where X is the input vector; W, H, and V are the weight matrices of the first hidden, second hidden, and output layers, respectively; N, J, K, are the numbers of nodes of the input, first hidden, and second hidden and layers, respectively; and b, β, and 𝒷 are the biases of the first hidden, second hidden, and output layers, respectively. The number of nodes in the output layer is I = 1 as a scalar is needed to estimate the input frequency. Bold type was used for matrices and vector functions, and plain-text symbols for scalar functions and variables.

The activation functions handle the relevant vectors pointwise as shown in Figure 1, while output variable f is a scalar that represents the input frequency.

The minimal number of hidden layers L for the successful extraction of input features is L = 2. A larger L may slightly improve performance, but it increases complexity.

### 5.2. Training Data

Training data were samples that were simulated using the model in Equation (3) on MATLAB (under Academic License 40635944). The number and magnitude of these samples are dependent on signal duration T, sampling frequency $f_{s}$, and the signal-to-noise ratio (SNR).

Data samples for each T and SNR were divided into three groups: 70% for training, 15% for testing, and 15% for validation.

The training function of the DL network was chosen to be the MATLAB trainscg function, which updates weight and bias values according to the scaled conjugate gradient method, with learning rate of $L_{r}\;=\;0.1$ and number of epochs p = 100.

As changes in carrier frequency are expected to be relatively small (mostly due to the Doppler effect [19]), a small low-frequency Hz range [20,21,22] was considered in this experiment, with a sampling frequency of $T_{s}\;=\;0.01$ Hz. This makes normalized frequency $f/f_{s}$ be in the range of [0.23, 0.25]. This 9is the normalized frequency that is considered while handling a frequency-estimation problem [20]. On the other hand, small ranges of small frequencies are expected in power systems (e.g., 50 Hz) and biosystems (for example, electroencephalograms (EEG), in the range of 0–40 Hz) [21,22].

Higher ranges are possible but require more training samples N and thereby more complex networks; for example, handling a maximal frequency of 100kHz may require N = 2000 input nodes [9].

The necessary number of input samples that affects DL performance is related to number γ of complete periods (cycles) of the input sinusoid that are included in the training process, which can be expressed as
(15)$$\gamma \;=\;\frac{T}{T_{o}}\;=\;\frac{T/T_{s}}{T_{o}/T_{s}}\;=\;\frac{N}{\nu_{o}},$$
where $\nu_{o}$ is the number of samples in one cycle of the input signal with frequency $f_{o}$ and period $T_{o}$, given by
(16)$$\nu_{o}\;=\;\frac{T_{o}}{T_{s}}\;=\;\frac{f_{s}}{f_{o}}\;,$$
from which the number of cycles used in [9], which used $T_{s}\;=\;1\;\mu s$ (hence, $f_{s}\;=\;1\times 10^{6}$ Hz) and a maximal input frequency of $f_{o}\;=\;1\times 10^{5}$ Hz, can be calculated as
$$\gamma \;=\;\frac{N}{\nu_{o}}\;=\;N\frac{f_{o}}{f_{s}}\;=\;2000\;\frac{1\times 10^{5}}{1\times 10^{6}}\;=\;200.$$

In this work, the number of input cycles used to train the DL reached the limit of
$$\gamma \;=\;N\frac{f_{o}}{f_{s}}\;=\;11\frac{25}{100}\;=\;2.75,$$
giving reasonable results with almost 3 input cycles, as shown in Section 6.

Hence, the use of N = 2000 in [9] is an overestimation of the number of input samples that are necessary to train the DL network. While this overestimation may give the advantage of minor improvement in error performance, it is at the expense of increasing network complexity, which may not be attractive for resource-limited systems such as sensors in a wireless sensor network (WSN).

Generally, the choice of the proper number of input samples should be based on the value of γ, and this process represents a trade-off between complexity and accuracy.

The selected frequency range was divided with a frequency step of df = δ/μ, with δ = 25−23 = 2 and μ = 40 in the simulations.

Of course, more accuracy can be obtained with a larger μ, but this can be at the expense of more nodes in the network.

This frequency step is time-independent; hence, it is a design problem, unlike frequency step Δf = 1/T in DFT-based techniques, which is time-dependent and follows the Heisenberg uncertainty principle in the time–frequency domain [23].

For each incremental frequency $\left\{f_{i}\;=\;23+i\cdot df|i\;=\;0:\mu \right\}$ in the selected frequency range, a number of realizations R = 100 is generated, so that the training vector is
(17)$$X_{t}\;=\;{\left[S_{0}\;S_{2}\;\cdots \;S_{\mu }\right]}_{N\times \left(\mu +1\right)R};\;S_{i}\;=\;{\left[s_{i1}\;s_{i2}\;\cdots \;s_{iR}\right]}_{N\times R}$$
with labels
(18)$$L_{t}\;=\;{\left[L_{0}\;L_{2}\;\cdots \;L_{\mu }\right]}_{1\times \left(\mu +1\right)R};\;L_{i}\;=\;i*{\left[1\;1\;\cdots \;1\right]}_{1\times R},$$
where $s_{ik}$ is the N×1 column that represents the kth realization of Equation (3) with a frequency $f_{i}$ under the same SNR. Hence, there is a total of μ+1 classes. A larger number of realizations R during the training phase implies better training.

### 5.3. Training Function: Scaled Conjugate Gradient Backpropagation

The training function is the most important part of network design as it is responsible for finding the proper weight and bias values. Several training functions were tried for the FE problem in this work, but most either failed or gave a large error (insufficient performance). The best function for FE problems is the scaled conjugate gradient (SCG) backpropagation algorithm [24,25].

SCG’s superior performance is because it user-independently updates its parameters at each iteration [24]. In addition, SCG uses multiple adaptivity where the step size and search direction are updated as shown in Equations (29) and (30). Several applications confirmed this (see, for example, [26]).

The SCG optimization function combines the Levenberg–Marquardt (LM) algorithm (which is, in turn, a combination of gradient descent and the classical Gauss–Newton algorithm) with the conjugate-gradient approach [24]. The training process utilizes iterative algorithms to update the weights and biases of the network based on multivariate error function
(19)$$e\left(v\right)\;=\;\frac{1}{2}\sum_{k\;=\;1}^{R}e_{k}{}^{2}\;=\;\frac{1}{2}\sum_{k\;=\;1}^{R}{\left(f_{k}-f_{o}\right)}^{2},$$
where $f_{k}$ is the output of the network at the kth realization. All training algorithms minimize this global error by the optimal adjustment of multivariate
(20)$$v\;=\;{\left[w\;b\right]}_{1\times \xi },$$
which includes weights (w) and biases (b) with ξ/2 = number of nodes.

The update rule of the LM algorithm is given at the ith iteration (epoch) by
(21)$$v_{i+1}\;=\;v_{i}-{\left[\nabla^{2}e\left(v_{i}\right)+L_{r}I\right]}^{-1}\nabla e\left(v_{i}\right),$$
where ∇e is the gradient operator, damping parameter $L_{r}$ controls the update process (the learning rate), and I is the identity matrix. The Hessian matrix is approximated by
(22)$$\nabla^{2}e\left(v_{i}\right)\approx J^{T}J,$$
while the gradient can be written as
(23)$$g_{i}\;=\;\nabla e\left(v_{i}\right)\;=\;J^{T}e\left(v_{i}\right),$$
where
(24)$$J\;=\;{\left[\frac{\partial e_{k}}{\partial v_{j}}\right]}_{R\times \xi }$$
is the Jacobian matrix. Hence, LM weight-bias update can be written as follows.
(25)$$v_{i+1}\;=\;v_{i}-{\left[J^{T}J+L_{r}I\right]}^{-1}J^{T}e\left(v_{i}\right).$$

The optimal solution is v, which makes g = ∇e(v) = 0, and $v_{o}$ is random.

Learning rate $L_{r}$ balances the performance between Newton’s method (when it is zero) and the gradient descent (when it is large).

In the first iteration (epoch) of the SCG, $v_{o}$ is chosen at random. Then, LM is used to find the steepest descent, which is the negative of LM gradient
(26)$$p_{o}\;=\;-g_{o}.$$

Then, optimal step $\alpha_{k}$ [26] is computed as follows.
(27)$$\alpha_{k}=-\frac{\nu_{k}^{T}\rho_{k}}{\nu_{k}^{T}G_{k}},$$
(28)$$G_{k}\;=\;\underset{\epsilon \to 0}{\lim}\frac{g_{k}\left(v_{k}+\epsilon \cdot v\right)-g_{k}\left(v_{k}-\epsilon \cdot v\right)}{2\epsilon },$$
with *v* = arbitrary ξ-dimensional vector and ϵ = a small positive number. The two gradients $g_{k}\left(v_{k}+\epsilon \cdot v\right)$ and $g_{k}\left(v_{k}-\epsilon \cdot v\right)$ are evaluated by LM. The new weight-bias vector is given by

*ν_k+1_* = *ν_k_ + α_k_ρ_k_*,.(29)

Then, gradient $g_{k+1}$ is evaluated by LM at $v_{k+1}$.

Now, a new search direction is conjugate to the previous one using the following relation.
(30)$$p_{k+1}\;=\;-g_{k+1}+\beta_{k}p_{k},$$
where parameter $\beta_{k}$ is given by
(31)$$\beta_{k}\;=\;\frac{g_{k+1}^{T}g_{k+1}}{g_{k}^{T}g_{k}},$$
which is known as the Fletcher–Reeves formula [27]. Then, the update continues by going back to Equation (27) to find new step $\alpha_{k+1}$.

## 6. Results: Deep Learning vs. Classical Single-Tone FE

In this section, we analyze the performance of the DL approach for the frequency estimation of single tones in comparison with computationally efficient classical techniques.

For each SNR, a different network is trained to recognize frequencies in the specified frequency range. As the SNR may change in communication systems, this result may imply that a complicated set of DL-based estimators is necessary for proper functioning, and this complexity is not necessary.

The number of signal samples N used in the design of the estimation system has significant impact on FE performance in both DFT- and DL-based techniques. Therefore, a larger signal duration T (allowed for by the estimation system design) gives a larger $N\;=\;T/T_{s}$, and thereby better estimation.

Even better results can be obtained via more realizations. During the training phase, a larger number of realizations r implies better training. In the testing phase (actual working environment), obtaining a number ρ of different realizations is possible via dividing the input samples into sets of specific length that equal signal length N, which was selected during training (i.e., the number of input nodes to the trained DL network). Hence, for better frequency estimation, the transmitter should initially send the carrier single tone for sufficient sending duration P that includes a multiple ρ of training duration T (of which the sample length is $N\;=\;T/T_{s}$), i.e., P = ρT or $\rho N\;=\;P/T_{s}$. The final frequency estimate is the average of estimates over ρ realizations.

### 6.1. Performance under Different Input Lengths

First, we compared the performance of the DL-based FE with the selected classical DFT-based techniques for different values of the training input size (N). Figure 2 shows the performance of DL method with J = K = 5 versus selected classical techniques. The sinusoidal frequency was selected to be $f_{o}\;=\;24.41$ Hz. For a small N, the classical DFT-based approach failed to correctly estimate the input frequency.

The number of realizations during the training phase was selected to be R = 100. Considering different realizations in the training phase improves the network design, so that it would be ready to handle different situations in the application phase. During the application (working) phase, the number of realizations was selected to be ρ = 30. These working realizations improve estimation performance. In this paper, network performance was considered for ρ = 30. This choice of ρ is user-dependent and has no effect on network complexity.

Figure 3 shows the performance of the DL-based approach versus the best of the selected DFT-based approaches, the barycentric approach, for different input lengths, while Figure 4 shows the performance of the DL-based approach versus the barycentric approach with fewer nodes in the network, such that J = K = 3.

No significant difference was seen in error performance, as shown in Figure 4 (for J = K = 3), when compared with the error performance shown in Figure 3 (for J = K = 5). Despite the reduction in the number of nodes in both hidden layers of the network from 5 to 3, the error curves had almost the same trajectories versus SNR.

### 6.2. Performance under Insufficient SNR Information

SNR estimation is necessary for many purposes in communication systems, and it is needed for DL frequency estimation, as training is performed under a specific SNR. However, in some systems (e.g., some sensor networks), limited computational capabilities may not allow for SNR estimation except probably for a rough estimate. Knowing the exact value of SNR did not cause the performance of DL estimation to deteriorate, where only minor changes in the performance were noticed, as shown in Figure 5. Hence, the same network could be used for FE under different SNRs.

Relative deviation ν of DL performance for FE, averaged over M values of SNR, is defined as
(32)$$\nu \;=\;\frac{1}{M}\sum_{k\;=\;1}^{M}\frac{e_{b}\left(k\right)}{e_{o}\left(k\right)},\;$$
where $e_{o}\left(k\right)$ and $e_{b}\left(k\right)$ are the average errors at the kth SNR value (dB) under normal operation (correct SNR information) and biased SNR, respectively. Figure 6 shows relative deviation when J = K = 5. Nonsymmetric deviation was expected for positive and negative bias due to the nonsymmetry of the FE error performance curves (see Figure 2, Figure 3 and Figure 4).

### 6.3. Effect of Increasing Nodes at Hidden Layers

Generally, no significant performance difference was found when increasing the number of nodes in the hidden layers, as shown in Figure 7, where the performance comparison versus the barycentric method is presented using DLs with J = K = 5 and J = K = 3.

### 6.4. Impact on IoT, Sensors, and SDR

Reduced complexity can lead to the efficient implementation of FE, since DL does not need general-purpose digital multipliers if implemented as a multibit system, where trained weights can be implemented using look-up tables.

On the other hand, cheap and hardware-efficient implementation using short word length (SWL) is possible due to reduced complexity, where complicated digital multipliers are replaced by simple multiplexers.

Another hardware gain is because DFT techniques require the use of a complex-valued arithmetic, while DL systems use only a real-valued arithmetic. The implementation of a large DFT is a very complicated task for both hardware and software designs despite the significant efficiency provided via the latest versions of the fast Fourier transform (FFT) algorithm, which reduces the number of (complex) multiplications from $O\left\{N^{2}\right\}$ to $O\left\{N\log_{2}\left(N\right)\right\}$ [28].

From this work and [9,10,11], it can be inferred that N is proportional to input frequency $f_{o}$; hence, for high frequencies, size N can be large if DL is used, and much larger for classical FE if the same accuracy as that of DL is required, calling for very complicated DFT implementation in the complex-valued domain.

FFT implementation via parallel-computing algorithms can alleviate the time cost [20]; however, this approach is very expensive, power-consuming, and hardware-costly, which may not be suitable for many systems, especially in emerging IoT sensor applications.

Large-scale sensor applications require tiny, low-power, low-cost implementation, such as sensors used for healthcare monitoring, the early detection and monitoring of forest fires, natural-disaster detection, and large-scale surveillance for security applications.

Complicated, complex-valued operations increase both system size and complexity, and power consumption, which is a crucial factor in sensors deployed in WSNs, as the lifetime of these sensors is limited by the irreplaceable battery life [18].

For DL-based approaches, the number of (real) multiplications is mostly at the input stage, which is N, while other node multiplications are normally as low as 6 real-valued multiplications (which are trivial operations as they use look-up tables instead of digital multipliers); hence, the overall number of real-valued multiplications is O{N}.

Whether implemented using SWL or multibit technologies, this reduced complexity (versus the complicated, complex-valued classical FE techniques) makes the DL-based approach the most suitable FE technology in sensors and sensor networks, which are inherently linked to modern IoT applications.

Similar gains are obtained if implementation is performed via software, where reduced complexity leads to significant time gain, a factor that is attractive for software-defined radio (SDR) communications.

### 6.5. Future Directions: DL in the SWL Domain

The authors are currently investigating the possible implementation of the DL-based FE using short word length (SWL) where single-bit and ternary technologies are considered.

As the training phase can be performed offline, training could be accomplished using multibit computer simulation; however, training data should be converted into the SWL format. In this case, the new factor of quantization noise emerges, and this can be handled by better noise shaping via the proper selection of oversampling ratio (OSR), the quantization process, and the filter design of the sigma–delta modulator (SDM).

As adaptivity is now performed offline, this DL design for SWL could be even less complicated than the design of adaptivity in SWL systems is, first introduced in [29] as a result of SWL research supported by the Australian Research Council via discovery grant DP0557429.

The successful SWL implementation of DL-based FE could transform many other resource-limited IoT and sensor applications after the advent of DL techniques into the SWL domain.

## 7. Conclusions

This paper presented comprehensive analysis of the performance of a deep-learning based approach for the frequency estimation of a single-tone sinusoid. This work fills in the gaps of unhandled issues in recently proposed methods based on deep learning. The paper presented analysis of the frequency-estimation error under a range of signal-to-noise ratios (SNRs). Hidden layers can be as few as two, each with two or three nodes. The number of input samples (nodes) can be very small as compared to that in DFT-based classical methods, with much better error performance. In addition, performance was not significantly degraded in the case of wrong information about the actual value of the SNR. These results can reduce the complexity, power consumption, and cost of the communication system, properties that can be very useful for systems with limited memory and computational capabilities, such as wireless sensor networks (WSNs), which are inherently linked to current and future applications on the Internet of Things. Reduced complexity without using a complex-valued arithmetic also makes the DL technology for frequency estimation suitable for software-defined radio. Exploiting this reduced complexity to introduce DL in the short-word-length (SWL) domain was presented as a future direction.

## Figures and Tables

**Figure 1 sensors-21-02729-f001:**
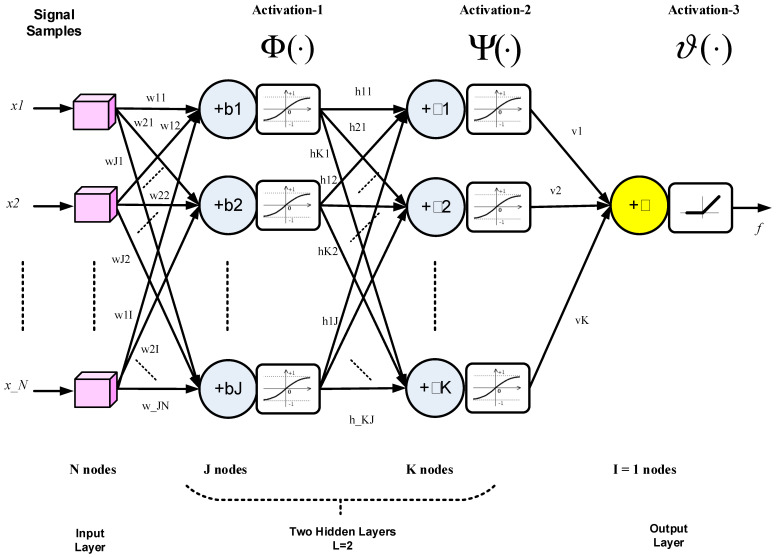
Structure of two-layer network for deep-learning (DL)-based frequency estimation (FE).

**Figure 2 sensors-21-02729-f002:**
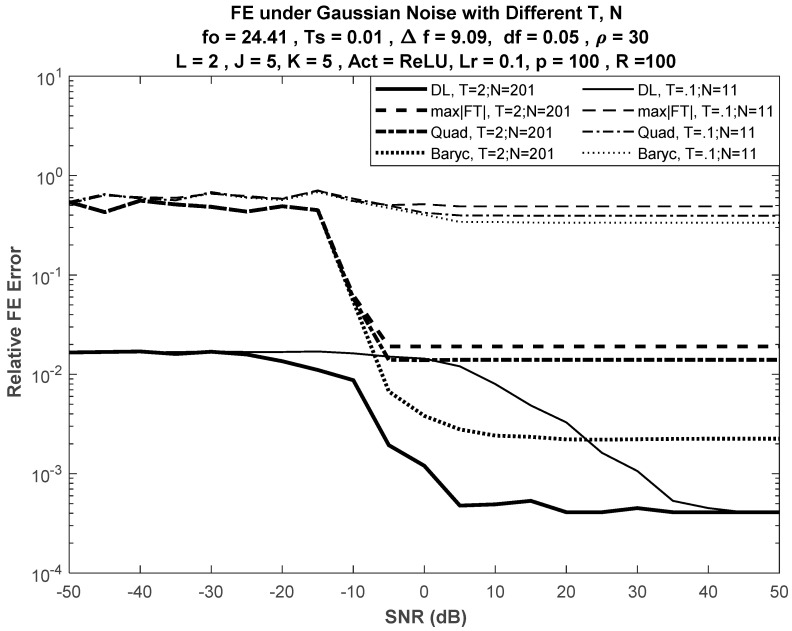
Performance of FE methods for different input lengths. Bold curves are for larger duration T = 2 s, N = 201 input samples; plain curves are for smaller duration T = 0.1 s, *N* = 11.

**Figure 3 sensors-21-02729-f003:**
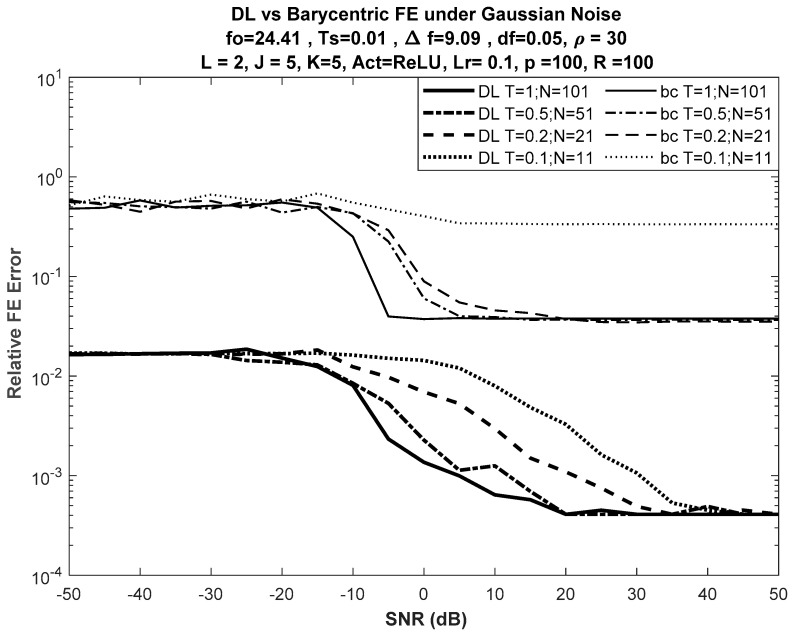
DL vs. barycentric discrete Fourier transform (DFT)-based FE for different input lengths with J = K = 5. Note that DFT-based FE failed when N = 11, while DL-based FE gave reasonable results even at low signal-to-noise ratio (SNR).

**Figure 4 sensors-21-02729-f004:**
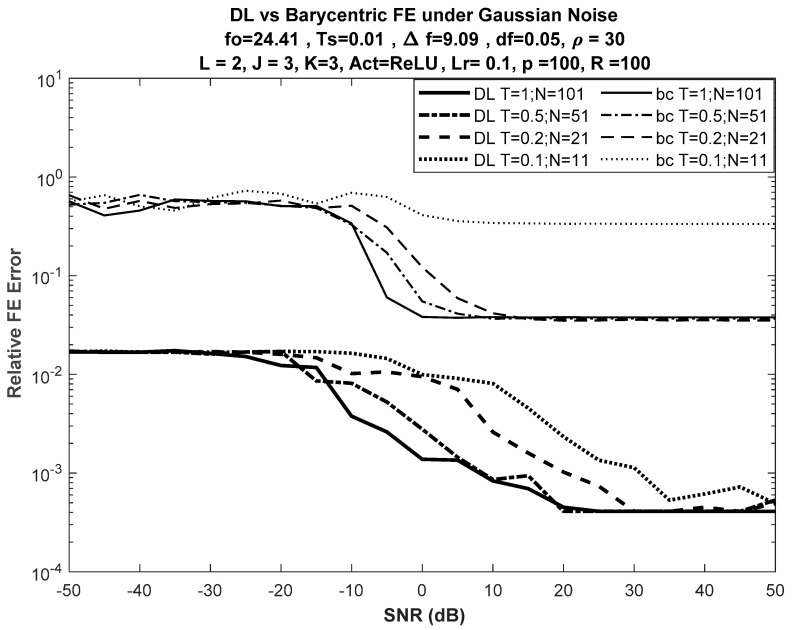
DL vs. barycentric DFT-based FE for different input lengths with J = K = 3. Bold curves used for DL.

**Figure 5 sensors-21-02729-f005:**
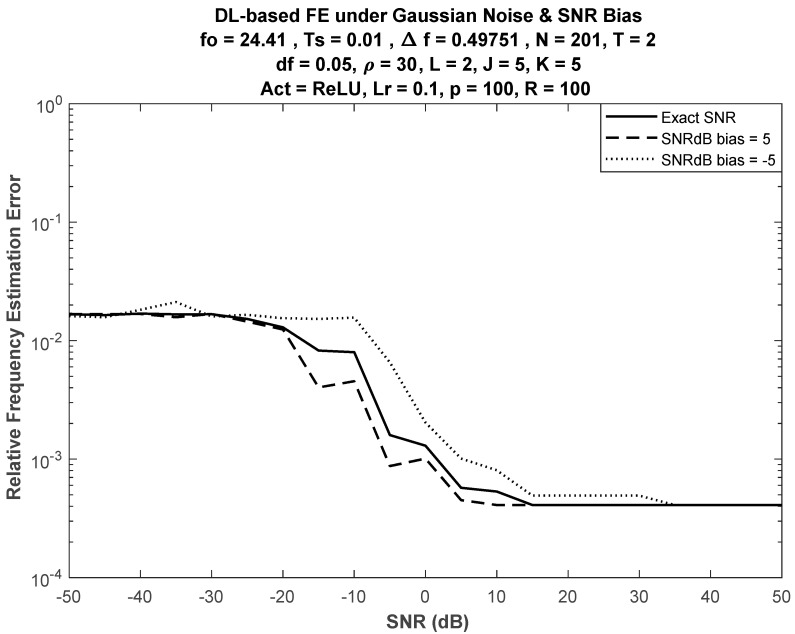
DL-based FE under bias from training SNR with J = K = 5.

**Figure 6 sensors-21-02729-f006:**
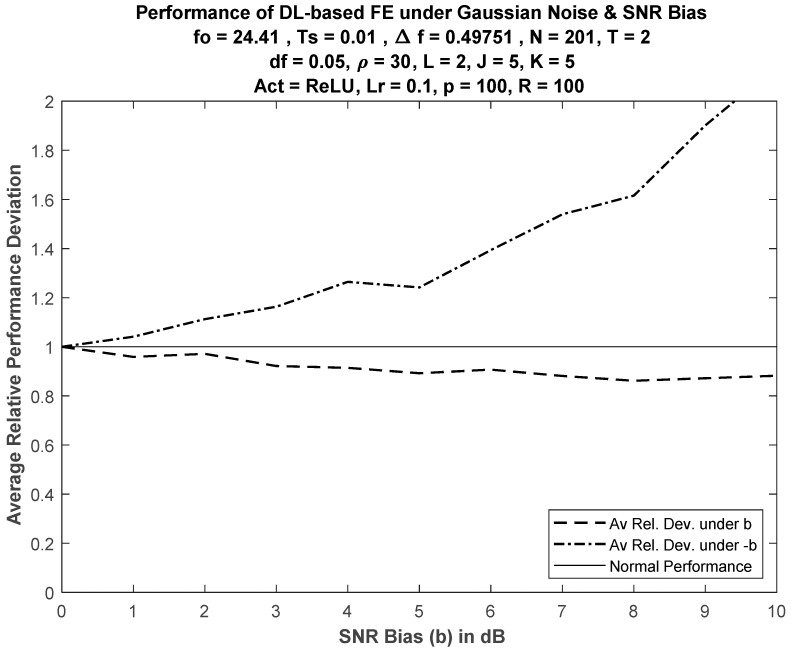
Average performance deviation of DL-based FE versus SNR bias.

**Figure 7 sensors-21-02729-f007:**
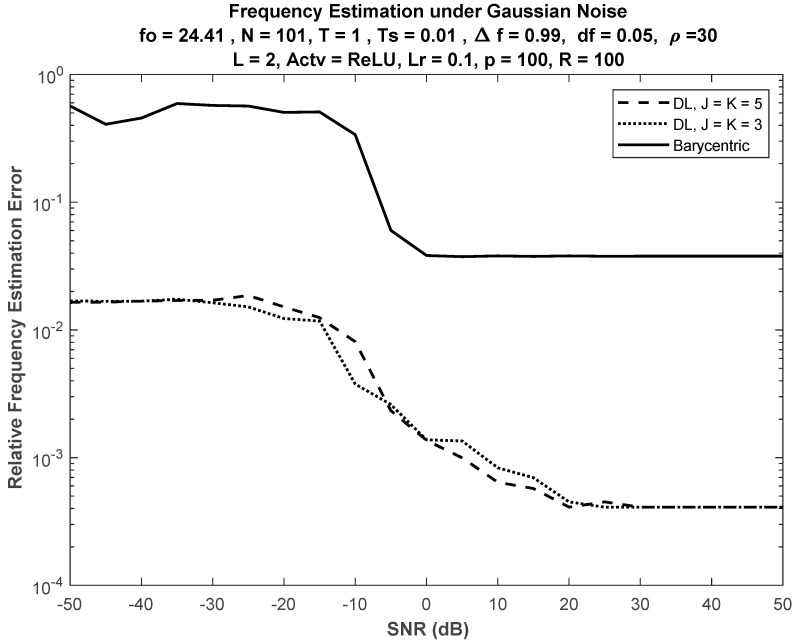
Performance of DL-based FE under different sizes of hidden layers. No significant difference in DL performance between networks with sizes J = K = 5 and J = K = 3.

## Data Availability

All types of data were generated using mathematical equations.

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
