# Peer review of "Deep Learning versus Spectral Techniques for Frequency Estimation of Single Tones: Reduced Complexity for Software-Defined Radio and IoT Sensor Communications"

_sensors, 2021, doi:10.3390/s21082729_

Round 1
Reviewer 1 Report
The paper has proposed a novel frequency estimation technique for single tone-based sinusoidal wave using deep learning. Since the deep learning has been applied in many areas of wireless communication with improved performance, the paper has contribution to some extent. However, the paper has many problems, which are listed as follows. The authors should justify the contribution of this work with respect to [9]. The organization of this paper is not that. The presentation of technical content in each section can be improved significantly. For example, in the problem formulation, the authors have highlighted the technical points in an enumerated form which should be in the contribution part at the beginning of the paper. Although deep learning-based frequency estimation is the main contribution of this paper, the technical content about this topic is provided in Section 5 with only 2 pages, which is very weird for a technical contribution. The authors should provide more information about the solution approach of this paper. Overall, the organization and presentation. The authors are required to work more in this context. All acronyms must be defined correctly. At the first usage, these should be defined and then could be used in the subsequent discussions. Putting semi-colon at the middle of any sentence is not a right approach for technical papers. The writing of this paper can be improved significantly. There are many grammatical, typographican and sentence construction errors here and there throughout the paper. In the following, I have highlighted some errors. Please fix the remaining paper following the similar strategy. page 1, line 11: It is shown that DL network with two layers having a few nodes can estimate frequency more accurately than well-known classical techniques. =====> It is shown that DL network with two layers having a few nodes can estimate frequency more accurately compared to well-known classical techniques. page 1, line 13: The study is comprehensive, filling gaps of existing works, where it analyzes error under different signal-to-noise ratios, numbers of nodes, and numbers of input samples; also, under missing SNR information. =====> While filling the gap in the existing literature, the study is comprehensive, which analyzes error under different signal-to-noise ratios (SNR), numbers of nodes, and numbers of input samples under missing SNR information. page 1, line 17: It is possible for DL to use as little as two layers with two or three nodes each, with complexity of O{N} versus O{Nlog2 (N)} for DFT-based FE, noting that less N is required for DL. =====> It is possible for DL to use as little as two layers while having two or three nodes for each, with complexity of O{N} compared with DFT-based FE with O{Nlog2 (N)} complexity. page 1, line 19: Hence, DL can significantly reduce FE complexity, memory, cost, and power consumption, making DL-based FE attractive for resource-limited systems like some IoT sensor applications. ======> Therefore, DL can significantly reduce FE complexity, memory cost and power consumption, which is attractive for resource-limited systems such as some IoT sensor applications. page 1, line 27: Estimating the frequency of a single-tone sinusoidal wave under noise has been a fundamental problem in signal processing for communications, and its effect extends to biomedical and power engineering [1,2]. =====> Estimating the frequency of a single-tone sinusoidal wave under noise has been a fundamental problem in signal processing for communications, and its effect is extended to biomedical and power engineering [1,2]. page 1, line 34: techniques; however, those techniques are far less accurate than Fourier-based techniques, especially under low values of signal-to-noise ratio (SNR) [4]. ======> techniques. However, those techniques are far less accurate compared to Fourier-based techniques, especially under low values of signal-to-noise ratio (SNR) [4]. page 1, line 36: Phase-locked loops (PLLs) are widely used in communication systems to handle this problem; however, PLLs can be slower that spectral or correlative techniques as they need time for locking [5]. =====> Please fix this sentence in the similar that I provided correction in the previous sentences. page 1, line 41: Recently machine learning has been emerging as a powerful tool for various tasks in many fields including communications and signal processing [6-8]. =====> Recently, machine learning has been emerging as a powerful tool for various tasks in many fields including communications and signal processing [6-8]. page 2, line 45: Unlike Fourier-based, PLL, or correlative techniques, DL-based approach requires the prior knowledge of SNR; however, SNR estimation can be in error under varying noise conditions, therefore many studies, including [9], assume that SNR is constant during the period of frequency estimation. =====> Please fix The Xlabel, Ylabel, legend and caption of all figures need to be improved and consistent. For example, Xlabel SNR(dB) should be SNR (dB). Ylabel "Relative FE error" should be "Relative FE Error"
Reviewer 2 Report
These comments mainly address the clarity of the paper for the reader noting that some of these points raised do become clear with eventual re-reading of the paper several times.
P.3 line 96
…the amplitude normally has less frequency content than the sinusoidal frequency…
Is a bit clumsy – what you mean is typically the time-varying amplitude a(t) is typically lower bandwidth or of a lower frequency “much lower” or << f0 – the sinusoidal carrier frequency is the model usually applied
p.3 line 104
can you give examples – is this meant to be in specific systems where frequency estimation is the first step and a pure pilot carrier is sent? This is not invariably the case for communication systems, certainly ‘acquisition’ is a step but is implemented in lots of different ways.
Also Doppler effect only applies in certain situations e.g. fast moving mobile devices, it would not apply significantly to a geostationary satellite signal for example as long as the receiver wasn’t moving rapidly.
p.6 line 240
By a small range [23-25]Hz – I presume this means the carrier frequency itself is meant to be in the 20s of Hz? Or does that mean its size varies with a delta_f of 23-25 Hz? This notation is not particularly clear. The application area – biosensors/IoT – is now mentioned. What is the size of the Doppler effect expected? Perhaps it would help if the application area, typical carrier frequencies, Doppler effects etc. were explained.
It might be useful to add some discussion about realistic frequencies used in applications and the impact this might have on the proposed methods DL or otherwise - although that might be premature seeing as clearly more work is in progress on implementation aspects.
These comments are about the Results sections.
p.7 line 290
No significant difference is noticed as compared to Figure 3. – should this read as compared to Figure 2? No significant difference in what?
Is there a way to make the presentation of the figures a bit more concise e.g. especially combining Fig2a and 2b side by side.
Not sure it’s necessary to repeat in every figure label that the DFT methods apart from barycentric fail absolutely under certain conditions.
General comments
There could be greater development of the scenarios outlined in section 2 at the bottom of p.3 and the Results section. For example it is claimed that “
the possibility of different realizations in the working environment 142 (application phase) has not been handled previously. In this work we discuss 143 the effect of different realizations during the application phase. “ however this is not clearly signposted in the Results section. Is this the ‘different input lengths’? and/or different numbers of nodes in the hidden layers – although these are mentioned separately and individually in Section 1.
Round 2
Reviewer 1 Report
The paper is much improved in this version. However, still, there are some problems, which are listed as follows.
In the related work, while discussing the contribution of [9], a list of points were brought out which are shown in enumerated form with question marks. This is very weird. The authors should discuss the contribution of this work in regular writeup with active or passive voice (no question mark or enumeration)
Throughout the paper, the length of each paragraph is not that long, which is not elegant. Each paragraph should have reasonable size.
I still believe, the authors should provide more information about the solution approach (Section 5) using DL
The writing of this paper can be improved still significantly. In the following, I have listed some such problems. The authors are required to find rest of the similar problems and fix those.
page 2, line 59: In many situations SNR estimation can be inaccurate or unavailable. =====> In many situations, SNR estimation can be inaccurate or unavailable.
page 2, line 61: The DL-approach is SNR-deendent, hence an investigation of the system performance under various SNR’s is presented. =====> The DL-approach is SNR-dependent, and hence an investigation of the system performance under various SNR is presented.
page 2, line 63: It is expected that the more nodes in the DL approach the better is the accuracy of estimation. =====> It is expected that the more the nodes in the DL approach the better the accuracy of the estimation is.
page 2, line 74: The reduced complexity introduced by DL-based FE in addition to avoiding complex-valued arithmetic will make FE easier and cheaper for IoT communications, sensors, sensor networks, and software-defined radio (SDR). =====> The reduced complexity introduced by DL-based FE in addition to avoiding complex-valued arithmetic operation will make FE easier and cheaper for IoT communications, sensors, sensor networks, and software-defined radio (SDR).
In the entire paper, when an equation is referred, it is presented as "Equation (eqn-num)". However, these should be as either "Eq. eqn-num" or "(eqn-num)"
In Eq. 5, SNRdB can be written in a much better way
In all figures, top label has ";" punctuation. This should be avoided in technical writeup
The Xlabel of Figure 6 should be "SNR Bias (b) in dB"
After "as follows", there should be full-stop instead of ":"
Reviewer 2 Report
The responses and changes to the paper have clarified the meaning and made the work done more accessible to readers.
